# Synthesis and Photophysical Properties of a Series of Dimeric Indium Quinolinates

**DOI:** 10.3390/molecules26010034

**Published:** 2020-12-23

**Authors:** Sang Woo Kwak, Ju Hyun Hong, Sang Hoon Lee, Min Kim, Yongseog Chung, Kang Mun Lee, Youngjo Kim, Myung Hwan Park

**Affiliations:** 1Department of Chemistry, Chungbuk National University, Cheongju 28644, Korea; sangwoo814@hanmail.net (S.W.K.); lsanh94@gmail.com (S.H.L.); minkim@chungbuk.ac.kr (M.K.); yschung@chuungbuk.ac.kr (Y.C.); 2Department of Chemistry, Institute for Molecular Science and Fusion Technology, Kangwon National University, Chuncheon 24341, Korea; asqm2253@kangwon.ac.kr; 3Department of Chemistry Education, Chungbuk National University, Cheongju 28644, Korea

**Keywords:** indium quinolinates, charge transfer, quantum yield, radiative decay constant, non-radiative constant

## Abstract

A novel class of quinolinol-based dimeric indium complexes (**1**–**6**) was synthesized and characterized using ^1^H and ^13^C(^1^H) NMR spectroscopy and elemental analysis. Compounds **1**–**6** exhibited typical low-energy absorption bands assignable to quinolinol-centered π–π* charge transfer (CT) transition. The emission spectra of **1**–**6** exhibited slight bathochromic shifts with increasing solvent polarity (*p*-xylene < tetrahydrofuran (THF) < dichloromethane (DCM)). The emission bands also showed a gradual redshift, with an increase in the electron-donating effect of substituents at the C5 position of the quinoline groups. The absolute emission quantum yields (Φ_PL_) of compounds **1** (11.2% in THF and 17.2% in film) and **4** (17.8% in THF and 36.2% in film) with methyl substituents at the C5 position of the quinoline moieties were higher than those of the indium complexes with other substituents.

## 1. Introduction

The creation of tris(8-hydroxyquinolinato)aluminum (Alq_3_) by Tang and Van Slyke pioneered a new era of group 13-based organometallic luminescent materials that can be used in versatile optoelectronic fields [1]. Numerous efforts and approaches have been used to modulate the quinolinate ligands and expand their applications in organic light-emitting diodes (OLEDs) [2,3,4,5,6]. In this context, particular emphasis has been placed on the development of tris-incorporated metal complexes (Mq_3_). Owing to the ease of introducing various substituents at the C2 and C5 positions of the quinolinolate moiety, studies of various tris-organometallic complexes based on quinolinate derivatives have also been conducted [7,8]. These complexes are endowed with photophysical properties that originate from the control of the highest occupied molecular orbital (HOMO) and the lowest unoccupied molecular orbital (LUMO) energy levels. Specifically, the systematic variation in the substituents at the C5 position of the quinolinolate groups led to excellent optical properties such as emission-color tuning and enhanced quantum efficiencies [9,10,11,12,13,14,15,16,17,18]. However, most of the previous studies primarily focused on tris-complexes.

Recently, our group reported a series of quinolinol-based indium complexes in which the sequential introduction of quinolinate ligands to the indium center could control both the emission color and quantum efficiency (Figure 1) [19]. Importantly, the dimeric indium complex (InMeq_1_) with a quinolinate ligand exhibited the highest quantum efficiency (Φ_PL_ = 59% in the poly(methyl methacrylate) (PMMA) film) compared to all the indium luminophores reported to date.

In this study, we designed a series of dimeric indium quinolinates with different substituents (Me, Br, and Ph) at the C5 position of two types of quinolinate ligands (q and Meq) to prove the substitution effects for developing potential indium-based luminescent materials. The detailed synthetic procedures and optical properties of these complexes were investigated.

## 2. Materials and Methods

### 2.1. General Considerations

All manipulations were carried out under an inert N_2_ atmosphere using the standard Schlenk and glovebox techniques. All anhydrous-grade solvents (*n*-hexane, diethyl ether, and toluene) were purchased from Alfa Aesar (Ward Hill, MA, USA) and dried by passing them through an activated alumina column and storing them over activated molecular sieves (5 Å). The spectrophotometric-grade solvents (*p*-xylene, tetrahydrofuran (THF), dichloromethane (DCM), and acetonitrile (MeCN)) were used as received from Merck (Darmstadt, Germany). All the commercially available reagents (2-amino-4-bromophenol, 2-amino-4-methylphenol, and 2-amino-4-phenylphenol) were purchased from Alfa Aesar (Ward Hill, MA, USA)and used without any further purification. The trimethylindium (InMe_3_) was analogously prepared according to the literature [19,20,21,22,23]. Because InMe_3_ is highly reactive and pyrophoric, it should be stored in a glovebox and used carefully. The quinolinol compounds 5-bromoquinolin-8-ol (**2a**) [24], 5-methylquinolin-8-ol (**3a**) [25], 5-phenylquinolin-8-ol (**5a**) [26], 5-bromo-2-methylquinolin-8-ol (**6a**) [27], 2,5-dimethylquinolin-8-ol (**7a**) [28], and 5-phenyl-2-methylquinolin-8-ol (**8a**) [29] were synthesized using previously reported methods. The deuterated solvent (CDCl_3_) from Cambridge Isotope Laboratories (Tewksbury, MA, USA) was used after drying it over activated molecular sieves (5 Å). The NMR spectra were recorded on a Bruker Avance 400 spectrometer (400.13 MHz for ^1^H and 100.62 MHz for ^13^C) (Bruker Corporation, Billerica, MA, USA) at the laboratory’s ambient temperature. The chemical shifts are given in ppm and are referenced against external Me_4_Si (^1^H and ^13^C NMR). The elemental analyses were performed on an EA3000 spectrometer (Eurovector, Pavia, Italy) in the Central Laboratory of Kangwon National University. The UV−vis absorption and PL spectra were recorded on Jasco V-530 (Jasco, Easton, MD, USA) and Fluoromax-4P spectrophotometers (HORIBA, Edison, NJ, USA), respectively. The fluorescence decay lifetimes were measured using an FLS920 time-correlated single-photon-counting spectrometer (Edinburgh Instruments, Livingston, UK) in the Central Laboratory of Kangwon National University, which was equipped with a picosecond pulsed diode laser (EPL 375-ps) pulsed semiconductor diode laser as an excitation source and a microchannel plate photomultiplier tube (200−850 nm) as a detector at 298 K.

### 2.2. Synthesis of [5-methyl-8-quinolinolate In(III)–Me_2_]_2_
*(**1**)*

A toluene solution (10 mL) of InMe_3_ (0.080 g, 0.50 mmol) was added to a toluene solution (20 mL) of **1a** (0.088 g, 0.55 mmol) at room temperature. The reaction mixture was stirred for 12 h, and the insoluble parts were collected by filtration. The remained solid was washed with *n*-hexane (3 × 20 mL) and dried in vacuo to obtain **1** as a pale-yellow solid (0.090 g, 62%). ^1^H NMR (CDCl_3_): δ 8.51 (dd, *J* = 1.2 and 4.0 Hz, 2H), 8.34 (m, 2H), 7.47 (m, 2H), 7.26 (dd, *J* = 1.4 and 3.6 Hz, 2H), 6.94 (d, *J* = 6.8 Hz, 2H), 2.55 (s, 6H), −0.13 (s, 12H, In−CH_3_). ^13^C{^1^H} NMR (CDCl_3_): δ 156.43, 143.58, 140.17, 134.97, 129.57, 128.92, 120.86, 119.58, 112.44, 17.62, −5.49. Anal. Calcd for C_24_H_28_In_2_N_2_O_2_: C, 47.56; H, 4.66; N, 4.62. Found: C, 47.38; H, 4.59; N, 4.57.

### 2.3. Synthesis of [5-bromo-8-quinolinolate In(III)–Me_2_]_2_
*(**2**)*

This compound was prepared in a manner analogous to the synthesis of **1** using **2a** (0.12 g, 0.55 mmol). The desired compound **2** was obtained as a yellow solid (0.12 g, 65%). ^1^H NMR (CDCl_3_): δ 8.58 (dd, *J* = 1.0 and 4.2 Hz, 2H), 8.54 (m, 2H), 7.69 (d, *J* = 8.4 Hz, 2H), 7.56 (m, 2H), 6.91 (d, *J* = 6.8 Hz, 2H), −0.11 (s, 12H, In−CH_3_). ^13^C{^1^H} NMR (CDCl_3_): δ 158.10, 144.60, 140.72, 137.95, 132.82, 128.80, 122.43, 113.72, 104.63, −5.19. Anal. Calcd for C_22_H_22_Br_2_In_2_N_2_O_2_: C, 35.91; H, 3.01; N, 3.81. Found: C, 36.31; H, 3.06; N, 3.74.

### 2.4. Synthesis of [5-phenyl-8-quinolinolate In(III)–Me_2_]_2_
*(**3**)*

This compound was prepared in a manner analogous to the synthesis of **1** using **3a** (0.12 g, 0.55 mmol). The desired compound **3** was obtained as a yellow solid (0.11 g, 60%). ^1^H NMR (CDCl_3_): δ 8.23 (d, *J* = 6.4 Hz, 2H), 7.84 (d, *J* = 6.8 Hz, 2H), 7.46–7.42(m, 6H), 7.40–7.37 (m, 2H), 7.34–7.33 (m, 4H), 7.16 (dd, *J* = 1.2 and 4.2 Hz, 2H), 7.08 (d, *J* = 7.2 Hz, 2H), −0.10 (s, 12H, In−CH_3_). ^13^C{^1^H} NMR (CDCl_3_): δ 161.23, 153.28, 147.73. 143.85, 141.08, 135.94, 132.03, 131.93, 125.56, 121.17, 118.04, 116.85, 107.75, −5.06. Anal. Calcd for C_34_H_32_In_2_N_2_O_2_: C, 55.92; H, 4.42; N, 3.84. Found: C, 55.81; H, 4.41; N, 3.74.

### 2.5. Synthesis of [2-methyl-5-methyl-8-quinolinolate In(III)–Me_2_]_2_
*(**4**)*

A toluene solution (10 mL) of InMe_3_ (0.080 g, 0.50 mmol) was added to a toluene solution (20 mL) of **4a** (0.095 g, 0.55 mmol) at room temperature. The reaction mixture was stirred for 12 h, and the insoluble parts were collected by filtration. The remained solid was washed with diethyl ether (3 × 20 mL) and dried in vacuo to obtain **4** as a pale-yellow solid (0.098 g, 62%). ^1^H NMR (CDCl_3_): δ 8.56 (dd, *J* = 1.4 and 4.2 Hz, 2H), 7.53 (dd, *J* = 1.2 and 3.8 Hz, 2H), 7.33 (d, *J* = 7.2 Hz, 2H), 7.00 (d, *J* = 7.8 Hz, 2H), 2.86 (s, 6H), 2.59 (s, 6H), −0.18 (s, 12H, In−CH_3_). ^13^C{^1^H} NMR (CDCl_3_): δ 158.93, 156.78, 140.52, 135.33, 129.93, 129.27, 121.21, 119.94, 112.80, 24.13, 17.97, −5.13. Anal. Calcd for C_26_H_32_In_2_N_2_O_2_: C, 49.24; H, 5.09; N, 4.42. Found: C, 49.20; H, 5.00; N, 4.38.

### 2.6. Synthesis of [2-methyl-5-bromo-8-quinolinolate In(III)–Me_2_]_2_
*(**5**)*

This compound was prepared in a manner analogous to the synthesis of **4** using **5a** (0.095 g, 0.55 mmol). The desired compound **5** was obtained as a dark yellow solid (0.11 g, 58%). ^1^H NMR (CDCl_3_): δ 8.56 (dd, *J* = 1.4 and 4.0 Hz, 2H), 7.52 (m, 2H), 7.33 (dd, *J* = 1.6 and 3.4 Hz, 2H), 7.00 (d, *J* = 7.8 Hz, 2H), 2.60 (s, 6H), −0.08 (s, 12H, In−CH_3_). ^13^C{^1^H} NMR (CDCl_3_): δ 157.88, 154.39, 140.50, 137.74, 132.60, 128.58, 122.21, 113.51, 104.41, 24.39, −5.40. Anal. Calcd for C_24_H_26_Br_2_In_2_N_2_O_2_: C, 37.73; H, 3.43; N, 3.67. Found: C, 37.68; H, 3.49; N, 3.63.

### 2.7. Synthesis of [2-methyl-5-phenyl-8-quinolinolate In(III)–Me_2_]_2_
*(**6**)*

This compound was prepared in a manner analogous to the synthesis of **4** using **6a** (0.13 g, 0.55 mmol). The desired compound **6** was obtained as a yellow solid (0.13 g, 67%). ^1^H NMR (CDCl_3_): δ 8.25 (d, *J* = 7.2 Hz, 2H), 7.46–7.44 (m, 6H), 7.42–7.40 (m, 2H), 7.35 (dd, *J* = 1.4 and 3.2 Hz, 4H), 7.18–7.17 (m, 2H), 7.10 (d, *J* = 6.8 Hz, 2H), 2.73 (s, 6H, CH_3_), −0.07 (s, 12H, In−CH_3_). ^13^C{^1^H} NMR (CDCl_3_): δ 160.94, 158.71, 149.14, 146.74, 142.52, 141.72, 136.40, 132.14, 127.64, 127.22, 121.21, 119.99, 113.69, 25.21, −5.16. Anal. Calcd for C_36_H_36_In_2_N_2_O_2_: C, 57.02; H, 4.79; N, 3.69. Found: C, 56.98; H, 4.66; N, 3.59.

### 2.8. Cyclic Voltammetry

The cyclic voltammetry (CV) measurements were performed in a deoxygenized MeCN (0.5 mM) solution with a three-electrode cell configuration (platinum working and counter electrodes and an Ag/AgNO_3_ reference electrode (0.1 M in MeCN)) using an AUTOLAB/PGSTAT12 system at room temperature. Tetra-*n*-butylammonium hexafluorophosphate (*n*-Bu_4_PF_6_) in MeCN (0.1 M) was used as the supporting electrolyte. The redox potentials were investigated at a scan rate of 100 mV/s and determined with respect to the ferrocene/ferrocenium (Fc/Fc^+^) redox couple.

### 2.9. Photophysical Properties

The samples for the UV–vis absorption and photoluminescence (PL) measurements were prepared using degassed solvents (*p*-xylene, THF, and DCM) in 1 cm quartz cuvettes (50 μM) at 298 K. The absolute PL quantum yields (Φ_PL_) of indium complexes **1**–**6** in THF solution were obtained using a Horiba Fluoromax-4P spectrophotometer equipped with a 3.2 inch integrating sphere (HORIBA, Edison, NJ, USA) at 298 K. The fluorescence decay lifetimes (τ) were measured using a FLS920 fluorescence spectrophotometer (Edinburgh Instruments, Livingston, UK) in time-correlated single-photon-counting (TCSPC) mode with a picosecond pulsed diode laser (EPL 375-ps) as a light source and a microchannel plate photomultiplier tube (MCP-PMT, 200–850 nm) as a detector at room temperature.

## 3. Results and Discussion

### 3.1. Synthesis and Characterization

Scheme 1 shows the routes for the synthesis of dimeric quinoline-based indium complexes **1**–**6**, which were easily produced in moderate yields (58–67%) by the reaction of 1.1 equivalent of the corresponding quinolines (**1a**–**6a**) with InMe_3_ in toluene at room temperature. Based on previously reported results, all the complexes were expected to exist as dimeric species in solution [20]. All the complexes were found to possess good solubility in common organic solvents. The formation of **1**–**6** was confirmed by ^1^H and ^13^C{^1^H} NMR spectroscopy (Appendix A) and elemental analysis. In particular, specific singlet signals assignable to the In–Me bonds were clearly observed in both the ^1^H (ca. 0.1 ppm) and ^13^C(^1^H) NMR (ca. −5.0 ppm) spectra of all the indium complexes.

### 3.2. Photophysical and Electrochemical Properties

To examine the photophysical properties of the dimeric indium complexes, UV−vis absorption and PL experiments were performed (Figure 2 and Table 1) in a diluted THF (50 μM) solution at 298 K. All the complexes **1**–**6** exhibited typical low-energy absorption bands in the range of 380 to 406 nm. The absorption bands can be ascribed to the quinolinol-centered π−π* charge transfer (CT) transition. The absorption maximum (λ_abs_) of these complexes gradually redshifted on increasing the electron-donating ability of the substituents at the C5 position of the quinolinate groups. The emission spectra of **1**–**6** displayed broad peaks in the range of 507 (green) to 523 (yellow) nm in THF, corresponding to a typical CT transition. The emission bands featured a gradual redshift phenomenon with an increase in the electron-donating effect of the substituents at the C5 position of the quinoline group (Figure 2 and Table 1). These results are not well-matched with the Hammett σ constants [30]. However, the observation indicated that the introduction of substituents with a high electron-donating effect at the C5 position of the quinolinolate group caused an increase in the HOMO energy levels in all the indium complexes. Furthermore, the emission maxima (λ_em_) of **1**–**6** exhibited slight bathochromic shifts in response to an increase in solvent polarity (*p*-xylene < THF < DCM) (Table 1; Appendix A). Such emission behavior indicated that all the dimeric indium complexes possessed polarized excited states. The solvatochromic nature of the broad emission bands confirmed that the PL spectra of compounds **1**–**6** correspond to the quinoline-based intramolecular charge transfer (ICT) transitions. The PL spectra of the compounds in the film (10 wt% doped with PMMA) displayed trends similar to those in the THF solution (Appendix A). The emission lifetime (τ) of **1**–**6** was measured to be in the range of nanoseconds in both the THF solution and the film state, indicating fluorescence (Table 1; Appendix A).

The absolute emission quantum yields (Φ_PL_) of these complexes were investigated in both the THF solution and the film state at room temperature (Table 1). The Φ_PL_ values of **1** (11.2% in THF and 17.2% in film) and **4** (17.8% in THF and 36.2% in film) with Me substituents at the C5 position of the quinoline moiety were determined to be higher than those of the indium complexes with other substituents in both the THF solution and the film state. The Φ_PL_ values gradually decreased as the electron-donating effect of substituents bound to the C5 position of the quinoline group increased (**1** → **2** → **3**: 11.2% → 6.6% → 0.05% in THF and 17.2% → 5.9% → 0.4% in film; **4** → **5** → **6**: 17.8% → 14.0% → 0.05% in THF and 36.2% → 18.0% → 0.4% in film). These results were elucidated by comparing the radiative decay constant (*k*_r_) with the non-radiative decay (*k*_nr_) constant for **1**–**6** in THF solution and the film state. As the electron-donating effect of C5 substituents increased, the *k*_r_ values gradually decreased (**1** (1.1 × 10^7^ s^−1^) > **2** (0.9 × 10^7^ s^−1^) > **3** (0.4 × 10^7^ s^−1^) in THF; **1** (1.2 × 10^7^ s^−1^) > **2** (0.9 × 10^7^ s^−1^) > **3** (0.9 × 10^7^ s^−1^) in film), while the *k*_nr_ values rapidly increased (**1** (8.6 × 10^7^ s^−1^) < **2** (13.3 × 10^7^ s^−1^ < **3** (719.6 × 10^7^ s^−1^) in THF; **1** (5.7 × 10^7^ s^−1^) < **2** (14.7 × 10^7^ s^−1^ < **3** (249.1 × 10^7^ s^−1^) in film). Importantly, the indium complexes **4**–**6** based on the Meq ligand possessed higher quantum efficiencies than those of the corresponding q ligand-based complexes **1**–**3**, similar to other dimeric indium quinolinates [20]. This feature is supported by the comparison of the *k*_r_ (**4** (1.8 × 10^7^ s^−1^) > **1** (1.2 × 10^7^ s^−1^), **5** (1.5 × 10^7^ s^−1^) > **2** (0.9 × 10^7^ s^−1^), and **6** (1.4 × 10^7^ s^−1^) > **3** (0.9 × 10^7^ s^−1^) in film) and *k*_nr_ values (**4** (3.2 × 10^7^ s^−1^) < **1** (5.7 × 10^7^ s^−1^), **5** (7.0 × 10^7^ s^−1^) < **2** (14.7 × 10^7^ s^−1^), and **6** (382.4 × 10^7^ s^−1^) < **3** (249.1 × 10^7^ s^−1^) in film) between the corresponding indium complexes in the THF solution and in the film state. These results imply that dimeric indium quinolinates based on the Meq ligand are more efficient luminophores.

Based on the electrochemical data obtained from CV measurements in MeCN, **1**–**6** showed totally irreversible oxidation processes (Figure 3 and Table 1). The HOMO energy levels and bandgaps (*E*_g_) of all the complexes were calculated using the measured onset oxidation potentials with the absorption edges (λ_abs_,_edge_). Contrary to the expectation, the calculated HOMO levels were found to decrease when the electron-donating effect of substituents at the C5 position of the quinoline groups increased. However, the calculated *E*_g_ values gradually decreased, which is consistent with the photophysical results.

## 4. Conclusions

In summary, we prepared a new series of quinolinol-based dimeric indium complexes (**1**–**6**) that exhibited low-energy absorption bands assignable to quinolinol-centered π–π* CT transition. The emission spectra of **1**–**6** exhibited a gradual redshift as the electron-donating effect of substituents at the C5 position of the quinoline groups increased. The quantum efficiencies of **1** and **4**, which had methyl substituents at the C5 position of the quinoline groups (q or Meq), were higher than those of the indium complexes with other substituents (Br and Ph). Consequently, these results provide a new perspective on the development of quinolinol-based dimeric indium complexes as potential organometallic luminophores. Further studies are underway to develop quinolinol-based indium complexes with improved quantum efficiencies for application as efficient luminescent materials.

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
