# Peer review of "Synthesis and Photophysical Properties of a Series of Dimeric Indium Quinolinates"

_molecules, 2020, doi:10.3390/molecules26010034_

Round 1
Reviewer 1 Report
In this manuscript the authors prepare and characterise a series on indium quinolate complexes and investigate the photophysical properties. The conclusion of the paper is that the methyl group in the 5 position enhances the quantum yields.
The paper is written well and the conclusions supported by the data presented by the authors. The science, whilst not spectacular, is propably of sufficient interest to the readers of Molecules to justify publication, subject to some minor comments below:
- the authors used 13C{1H} NMR spectroscopy so please change throughout the manuscript and esi
- The substituent effects could be linked to the physical organic properties of the molecules (hamett etc) if this data exists and maybe could then be used to predict what substituents would be even better for enhancing the QY?
- The ligands are rather common, so it would be interesting to report also other metal complexes that exist in the literature (e.g. Zn, Cd, Hg?) to compare and contrast the QY effects.
- Combining points 2 and 3 may then lead to a better understanding of the factors that really do influence the QY.
- Some idea of the errors associated with Table 1 would be most useful to determine if the trend the authors see is statistically significant or not.
Author Response
Response to reviewers’ comments
Reply to Reviewer 1
On behalf of the authors, I would like to thank you for reviewing our manuscript and for providing valuable suggestions. Our replies to the points you have raised are as follows and were considered in revising the manuscript.
Comments and Suggestions for Authors:
In this manuscript the authors prepare and characterise a series on indium quinolate complexes and investigate the photophysical properties. The conclusion of the paper is that the methyl group in the 5 position enhances the quantum yields.
The paper is written well and the conclusions supported by the data presented by the authors. The science, whilst not spectacular, is propably of sufficient interest to the readers of Molecules to justify publication, subject to some minor comments below:
1. the authors used 13C{1H} NMR spectroscopy so please change throughout the manuscript and esi
=> As the reviewer pointed out, we corrected them in the revised manuscript and supplementary material.
2. The substituent effects could be linked to the physical organic properties of the molecules (hamett etc) if this data exists and maybe could then be used to predict what substituents would be even better for enhancing the QY?
=> We are carefully convinced that the substituents of C2 position at the quinolate ligands could play a key role in enhancing the QY. In addition, it is expected that the electronic and steric effects of C2 substituents could affect the QYs of the complexes. Meanwhile, this work focused on the comparison of the C5 substitunts rather than the C2 substituents. Thus, we intend to proceed with further study in the next work.
3. The ligands are rather common, so it would be interesting to report also other metal complexes that exist in the literature (e.g. Zn, Cd, Hg?) to compare and contrast the QY effects.
=> Thank you for your valuable suggestions. It is likely that the study on group 12-based quinolate complexes would be interesting. To expand our research fields, we’ll attempt to further research topic.
4. Combining points 2 and 3 may then lead to a better understanding of the factors that really do influence the QY.
=> We fully agree with the reviewer’s comments. It is expected that the relationship between the substituents (C5 and C2 positions) and quantum efficiency will be more clearly understood by synthesizing compounds with more diverse substituents and looking at their photophysical properties. However, it is difficult to synthesize the quinolinate ligands various substituents. Nevertheless, we’ll do further study as the reviewer suggests. Thank you for your important comment again.
5. Some idea of the errors associated with Table 1 would be most useful to determine if the trend the authors see is statistically significant or not.
=> We fully agree with the reviewer’s comments.

Reviewer 2 Report
The manuscript by Lee, Kim, Park and coworkers describes the synthesis of a
series of dimeric indium quinolinates and their photophysical properties. A total of
six new indium complexes are prepared, that exhibited low-energy absorption
bands, and rather high quantum efficiencies, particularly those having methyl
substituents at C5 position of the quinoline groups. The authors have been
working recently in this field, and indeed, this manuscript could be considered as
an update of reference 20 (Inorg. Chem. 2019, 58, 8056). In general, the
manuscript is well written and enjoyable to read, and should be published at
Molecules. However, I have one major concern (see below) that should be
addressed before publication. Besides that, some typographical mistakes and other
comments are also pointed out below.
Major concern:
The authors state that: “…the absorption maximum (labs) of these complexes is
gradually red-shifted on increasing he electron-donating ability of the substituents at
the C-5 position of the quinolinate groups…”
And also, that:
“The FPL values gradually decreased as the electron-donating effect of substituents
bound to the C-5 position of the quinoline groups increased (i.e., 1 ® 2 ® 3:… 4 ® 5
® 6:…”
From these sentences it should be assumed that the electron-donating effect of the
substituents bound to the C-5 position of the quinoline group increase in the order:
Me < Br < Ph, which may be correct. However, this fact surprises me as according
to the Hammett s constants (Chem. Rev. 1991, 91, 165;
http://www.wiredchemist.com/data/hammett-sigma-constants), the electrondonating effect should increase in the order Br < Ph < Me. Of course, the effect at C5 position of quinolines may be different that the one expected from the known s
Hammett constants, and I am not aware that specific s Hammett constants have
been described for that position. Nevertheless, I think that this issue should be
mentioned in the manuscript and it may require a deeper discussion.
Typographical mistakes
Page 5, paragraph 2, lines 3 and 4. It is written: …absoprtion…; it should be
written: …absorption…
Materials and Methods:
Synthesis of 3. 1H NMR: Please, recheck the number of H atoms. I believe that, the
signal at 7.34-7.33 should integrate for 4H.
Synthesis of 5. 13C NMR: Please, recheck the peak list. I believe that there is an extra
peak listed.
Synthesis of 6. 1H NMR: Please, recheck the number of H atoms. 13C NMR: Please,
recheck the peak list. I also have some concerns regarding the identity/purity of
this compound as 14 aromatic signals can be seen in the 13C NMR spectrum and I
would only expect 13 aromatic signals.
References:
Ref 6, Please, include Jr. after last author last name as in refs. 3, 5, 12, 16.
Ref 13. Chem. Commun. did not have volume number in 2006. Please, remove issue
number.
Ref 19 = Ref 3.
Ref 24. Please, include the necessary brackets and tilde (written accent) for the last
name of first author.
Author Response
Response to reviewers’ comments
Reply to Reviewer 2
On behalf of the authors, I would like to thank you for reviewing our manuscript and for providing valuable suggestions. Our replies to the points you have raised are as follows and were considered in revising the manuscript.
Comments and Suggestions for Authors:
The manuscript by Lee, Kim, Park and coworkers describes the synthesis of a series of dimeric indium quinolinates and their photophysical properties. A total of six new indium complexes are prepared, that exhibited low-energy absorption bands, and rather high quantum efficiencies, particularly those having methyl substituents at C5 position of the quinoline groups. The authors have been working recently in this field, and indeed, this manuscript could be considered as an update of reference 20 (Inorg. Chem. 2019, 58, 8056). In general, the manuscript is well written and enjoyable to read, and should be published at Molecules. However, I have one major concern (see below) that should be addressed before publication. Besides that, some typographical mistakes and other comments are also pointed out below.
Major concern:
The authors state that: “…the absorption maximum (labs) of these complexes is gradually red-shifted on increasing he electron-donating ability of the substituents at the C-5 position of the quinolinate groups…”
And also, that:
“The FPL values gradually decreased as the electron-donating effect of substituents bound to the C-5 position of the quinoline groups increased (i.e., 1 ® 2 ® 3:… 4 ® 5 ® 6:…”)
From these sentences it should be assumed that the electron-donating effect of the substituents bound to the C-5 position of the quinoline group increase in the order: Me < Br < Ph, which may be correct. However, this fact surprises me as according to the Hammett s constants (Chem. Rev. 1991, 91, 165; http://www.wiredchemist.com/data/hammett-sigma-constants), the electron donating effect should increase in the order Br < Ph < Me. Of course, the effect at C5 position of quinolines may be different that the one expected from the known s Hammett constants, and I am not aware that specific s Hammett constants have been described for that position. Nevertheless, I think that this issue should be mentioned in the manuscript and it may require a deeper discussion.
=> Thank you for your valuable comments. As the reviewer mentioned, we also a little surprised at the result that are different from the Hammett σ constants presented by the references. And, we fully agree with the reviewer’s comment that “the effect at C5 position of quinolines may be different that the one expected from the known Hammett σ constants”. Nevertheless, we added the relevant sentences including Ref 30 in page 5 (line 166 and 167). Furthermore, we plan to conduct further studies closely related to present work.
Typographical mistakes
Page 5, paragraph 2, lines 3 and 4. It is written: …absoprtion…; it should be written: …absorption…
=> We corrected it in page 5 (line 160).
Materials and Methods:
Synthesis of 3. 1H NMR: Please, recheck the number of H atoms. I believe that, the signal at 7.34-7.33 should integrate for 4H.
=> We corrected it in page 2 (line 100).
Synthesis of 5. 13C NMR: Please, recheck the peak list. I believe that there is an extra peak listed.
=> We removed it.
Synthesis of 6. 1H NMR: Please, recheck the number of H atoms.
=> We corrected it in page 4 (line 123).
13C NMR: Please, recheck the peak list. I also have some concerns regarding the identity/purity of this compound as 14 aromatic signals can be seen in the 13C NMR spectrum and I would only expect 13 aromatic signals.
=> 13 Aromatic signals for 13C NMR of 6 are right. We revised it.
References:
Ref 6, Please, include Jr. after last author last name as in refs. 3, 5, 12, 16.
=> We corrected it in page 8 (line 249, 256, 270, and 280).
Ref 13. Chem. Commun. did not have volume number in 2006. Please, remove issue number.
=> We removed it.
Ref 19 = Ref 3.
=> We removed Ref 19 and overall, the references’ numbering was revised.
Ref 24. Please, include the necessary brackets and tilde (written accent) for the last name of first author.
=> We corrected it in page 9 (line 302).

Reviewer 3 Report
The scientific content of the ms is very interesting and this work deserves – according to my opinion – acceptance and publication in MOLECULES. I do believe that the paper will attract the interest of scientists working in the areas of quinolinol-based luminophores, the chemistry of indium(III) and the coordination chemistry of 8-hydroxyquinoline and its derivatives. Also, I am sure that the article will receive a respectable number of citations in the future. The general topic of quinolinol-based metal luminophores is currently “hot”. Since the exploitation of tris(8-hydroxyquinolinato)aluminium(III) as a green emitter in OLEDs, many research efforts have been made to discover more efficient coordination complexes based on quinolinate derivatives to widen their applicability in OLEDs. This work is a continuation of the previous project by the authors published in the ACS journal “Inorganic Chemistry”. Salient features of this work – which support my proposal for acceptance – are : (a) The emission spectra of the reported dinuclear In(III) complexes exhibit a gradual red-shift as the electron-donating effect of substituents at the C5 position of the quinoline group increases; and (b) The quantum efficiencies of two complexes having the methyl group at the C5 position is higher than those of similar In(III) complexes with other substituents at this position. The quality of the figures and schemes is good and the references list covers the topic under study satisfactorily. The Supplementary Materials section is very informative for the reader.
Based on the above mentioned, I am glad because I can propose acceptance of this fine piece of research in MOLECULES. I do not have scientific points to raise. Some comments/suggestions for the better organization of the paper are listed below:
- The formulae of the complexes, e.g. [In2(Me)4(1a-H)2], should appear somewhere in the ms. Simply writing 1, 2, 3, … creates confusion.
- It should be stated that the reported complexes are In(III) complexes, i.e. the metal is in the oxidation state III.
- The emission quantum yields of the reported dinuclear (and also mononuclear) analogous In(III) complexes.
- The figures’ numbering scheme is wrong and should be corrected; Figure 1 on page 6 should be numbered as Figure 2, and Figure 2 on page 7 should be numbered as Figure 3.
- Since the reported six complexes are new, I would welcome the incorporation of 3-4 figures with NMR and photophysical data in the main body of the ms; in the present form of the ms all the characterization data (Figs. S1-S13) appear in the Supplementary Info.
Author Response
Response to reviewers’ comments
Reply to Reviewer 3
On behalf of the authors, I would like to thank you for reviewing our manuscript and for providing valuable suggestions. Our replies to the points you have raised are as follows and were considered in revising the manuscript.
Comments and Suggestions for Authors:
The scientific content of the ms is very interesting and this work deserves – according to my opinion – acceptance and publication in MOLECULES. I do believe that the paper will attract the interest of scientists working in the areas of quinolinol-based luminophores, the chemistry of indium(III) and the coordination chemistry of 8-hydroxyquinoline and its derivatives. Also, I am sure that the article will receive a respectable number of citations in the future. The general topic of quinolinol-based metal luminophores is currently “hot”. Since the exploitation of tris(8-hydroxyquinolinato)aluminium(III) as a green emitter in OLEDs, many research efforts have been made to discover more efficient coordination complexes based on quinolinate derivatives to widen their applicability in OLEDs. This work is a continuation of the previous project by the authors published in the ACS journal “Inorganic Chemistry”. Salient features of this work – which support my proposal for acceptance – are : (a) The emission spectra of the reported dinuclear In(III) complexes exhibit a gradual red-shift as the electron-donating effect of substituents at the C5 position of the quinoline group increases; and (b) The quantum efficiencies of two complexes having the methyl group at the C5 position is higher than those of similar In(III) complexes with other substituents at this position. The quality of the figures and schemes is good and the references list covers the topic under study satisfactorily. The Supplementary Materials section is very informative for the reader.
=> Thank you for your excellent evaluation.
Based on the above mentioned, I am glad because I can propose acceptance of this fine piece of research in MOLECULES. I do not have scientific points to raise. Some comments/suggestions for the better organization of the paper are listed below:
The formulae of the complexes, e.g. [In2(Me)4(1a-H)2], should appear somewhere in the ms. Simply writing 1, 2, 3, … creates confusion.
It should be stated that the reported complexes are In(III) complexes, i.e. the metal is in the oxidation state III.
=> As the reviewer suggested, we added the formula of indium complexes (1–6) in page 3 (line 81, 90, 97, 104, and 113) and page 4 (line 120) of the revised manuscript.
The emission quantum yields of the reported dinuclear (and also mononuclear) analogous In(III) complexes.
=> The PLQYs of previously reported dimeric indium complexes, [8-quinolinolate In(III)–Me2]2 and [2-methyl-8-quinolinolate In(III)–Me2]2, were observed to be 9.7% and 22.8% in THF, respectively.
The figures’ numbering scheme is wrong and should be corrected; Figure 1 on page 6 should be numbered as Figure 2, and Figure 2 on page 7 should be numbered as Figure 3.
=> As the reviewer pointed out, we corrected them in page 5 (line 158 and 166), page 6 (line 201), and page 7 (line 209 and 216) of the revised manuscript.
Since the reported six complexes are new, I would welcome the incorporation of 3-4 figures with NMR and photophysical data in the main body of the ms; in the present form of the ms all the characterization data (Figs. S1-S13) appear in the Supplementary Info.
=> It seems to be okay in its current form without any problem.